# Linking Cerebrovascular Dysfunction to Age-Related Hearing Loss and Alzheimer’s Disease—Are Systemic Approaches for Diagnosis and Therapy Required?

**DOI:** 10.3390/biom12111717

**Published:** 2022-11-19

**Authors:** Carola Y. Förster, Sergey Shityakov, Verena Scheper, Thomas Lenarz

**Affiliations:** 1Department of Anaesthesiology, Intensive Care, Emergency and Pain Medicine, University of Würzburg, 97070 Würzburg, Germany; 2Laboratory of Chemoinformatics, Infochemistry Scientific Center, ITMO University, 197101 Saint-Petersburg, Russia; 3Department of Otorhinolaryngology, Hannover Medical School and Cluster of Excellence “Hearing4All”, 30625 Hannover, Germany

**Keywords:** Alzheimer’s disease, age-related hearing loss, neurovasculature, blood–brain barrier, blood–labyrinth barrier, spiral ganglion neuron, pharmacotherapy, neurotrophic factor, inner ear

## Abstract

Alzheimer’s disease (AD), the most common cause of dementia in the elderly, is a neurodegenerative disorder associated with neurovascular dysfunction, cognitive decline, and the accumulation of amyloid β peptide (Aβ) in the brain and tau-related lesions in neurons termed neurofibrillary tangles (NFTs). Aβ deposits and NFT formation are the central pathological hallmarks in AD brains, and the majority of AD cases have been shown to exhibit a complex combination of systemic comorbidities. While AD is the foremost common cause of dementia in the elderly, age-related hearing loss (ARHL) is the most predominant sensory deficit in the elderly. During aging, chronic inflammation and resulting endothelial dysfunction have been described and might be key contributors to AD; we discuss an intriguing possible link between inner ear strial microvascular pathology and blood–brain barrier pathology and present ARHL as a potentially modifiable and treatable risk factor for AD development. We present compelling evidence that ARHL might well be seen as an important risk factor in AD development: progressive hearing impairment, leading to social isolation, and its comorbidities, such as frailty, falls, and late-onset depression, link ARHL with cognitive decline and increased risk of dementia, rendering it tempting to speculate that ARHL might be a potential common molecular and pathological trigger for AD. Additionally, one could speculate that amyloid-beta might damage the blood–labyrinth barrier as it does to the blood–brain barrier, leading to ARHL pathology. Finally, there are options for the treatment of ARHL by targeted neurotrophic factor supplementation to the cochlea to improve cognitive outcomes; they can also prevent AD development and AD-related comorbidity in the future.

## 1. Introduction: Alzheimer’s Disease—Etiology and Pathophysiology

Alzheimer’s disease (AD) is a neurodegenerative disorder associated with neurovascular dysfunction [1], cognitive decline [2], and the accumulation of amyloid β peptide (Aβ) in the brain [3] and tau-related lesions in neurons termed neurofibrillary tangles (NFTs) [4]. AD is the most common cause of dementia in the elderly and is estimated to affect approximately 14 million people in the United States by 2050 [5]. While Aβ deposits and NFT formation are the central pathological hallmarks in AD brains, the majority of AD cases have been shown to exhibit a complex combination of multiple pathologies; as a result, the classification “systemic disease” might be appropriate [5].

Many hypotheses about the etiology and pathophysiology of AD have been developed in recent history over the past 20 years, the most prominent being centered around the above-mentioned abnormal deposit of Aβ protein and the formation of NFT. Respectively, general damage occurs to neurons through the effects of inflammation and oxidative stress. To introduce the topic for the purpose of this perspective, we will briefly summarize the current available knowledge on the pathophysiology of AD. For more details, please refer to in-depth reviews, e.g., Du et al. [6].

### 1.1. AD Pathophysiology

The amyloid hypothesis of AD claims that Aβ overproduction, toxicity and deposition initiate a sequence of events, culminating in neuronal injury and neuronal loss [7], that is considered the reason for cognitive decline [8]. Aβ dyshomeostasis is based on the deregulation of the amyloidogenic pathway with the downstream overproduction of Aβ (early-onset AD, primarily genetic-driven) and the failure of proteostasis from protein synthesis to protein degradation, with insufficient cerebral Aβ clearance (late-onset AD) [9].

In contrast, the tau hypothesis postulates that abnormalities of the microtubule-associated protein tau; it initiates the disease cascade as hyperphosphorylated tau has been shown to form NFTs within the cell bodies of affected neurons. This, in turn, leads to the disintegration of microtubules in brain cells. The destabilization of the microtubular system is speculated to disrupt the Golgi apparatus, in turn inducing abnormal protein processing and increasing the production of Aβ. Therefore NFT formation is regarded to result in the dysfunction of the biological activity between neurons and, later on, neuronal cell death [10].

The neurovascular hypothesis or so-called two-hit hypothesis of AD [11,12,13] suggests that cerebral Aβ accumulation is only the second insult (hit 2) in a sequence that is initiated by cerebrovascular damage (hit 1). Based on this hypothesis, the blood–brain barrier (BBB) is critical for brain Aβ homeostasis and regulates Aβ transport via the receptors LRP and RAGE [11]. BBB integrity is the primary reason for Aβ accumulation (compare with Section 2 “The Blood–Brain Barrier and its Dysfunction in the Genesis of AD”).

### 1.2. Noteworthy, Adding to This, Other Non-Aβ-Based Hypotheses Exist to Explain the Pathogenesis of AD

The cholinergic hypothesis claims that AD is caused by cholinergic effects such as the reduced synthesis of the neurotransmitter acetylcholine or the initiation of the large-scale aggregation of amyloid and neuroinflammation [14,15]. Most currently available drug therapies are based on this hypothesis [16].

Recent reports foster yet another theory: AD is a disease triggered by impaired APPs to establish a link between the loss metabolism, further promoted through tau pathology instead of Aβ amyloids; in short, very recently, a critical role for amyloid precursor protein (APP) and presenilin (PS) mutations in familial AD was highlighted, underlining that the trigger of AD might rather be linked to impairments of APP metabolism and an accumulation of APP C-terminal fragments. This is more significant than overproduction, failing the clearance and formation of Aβ itself. This rationale is strongly supported by the fact that all trials to treat AD by the usage of Aβ-targeted drugs have unexpectedly failed. For more details, please refer to, e.g., Kametani and Hasegawa (2018) [17,18,19].

Further hypotheses are based on the effects of environmental risk factors, such as smoking and infection, facilitating the development of AD.

A plethora of conditions have been listed as modifiable risk factors for AD development: lack of physical activity, sleep disturbances, dietary factors, smoking, and alcohol abuse but also cardiovascular disease, type 2 diabetes mellitus, epilepsy, brain trauma, anxiety and depression [20] and, importantly, age-related hearing loss (ARHL). Sensory changes (particularly the acquired disabilities of hearing and vision) and motor changes have been classified by the US National Institute on Ageing as so-called modifiable risk factors for AD. They have also been shown to contribute significantly to the cognitive symptoms of AD [21]. Modifiable risk factors are defined as treatable medical conditions and changeable lifestyle choices that have their own biological mechanisms while contributing to AD etiology and pathophysiology [20].

There is compelling evidence connecting ARHL with cognitive decline and an expanded risk of dementia. Indeed, ARHL, as a risk factor for AD, can confer an increased risk of social isolation in the elderly, which may, in turn, increase the risk of dementia. Hearing loss is associated with increased social isolation in older adults, likely through impaired communication. However, these two nosological forms can be connected directly as the association between early age-related hearing loss and brain β-amyloids [22]. Judging by this, we could speculate that amyloid-beta might damage the blood–labyrinth barrier (as it does to the BBB), leading to ARHL pathology.

Vascular damage is believed by many authors to play a crucial role in the pathogenesis of AD, as AD is a primarily vascular rather than neurodegenerative disorder [23]. This hypothesis is mainly based on the ability of cerebral hypoperfusion, caused by cortical microinfarcts, to exacerbate a cognitive decline in AD [24]. On the other hand, many epidemiological studies have shown the link between ARHL and AD associated with the loss of the brainstem and cerebellar volume [25]. Moreover, this biomarker has been established for various cardiovascular and cerebrovascular pathologies, including congenital heart disease, stroke and carotid stenosis [24,26,27].

In our review, we therefore present AD as a systemic malady that may be caused by ARHL-related vascular decline in BBB and BLB integrity and function and following damage to the CNS and spiral ganglion neurons. Our focus is to summarize key research on the establishment of a link between hearing loss and AD development by highlighting that (i) several illnesses and hazardous risk factors, such as systemic metabolic dysfunction, contribute to the ontogeny of AD and have been linked to risk of AD; (ii) the causes or impact relations between these conditions remain imprecisely defined; (iii) common constituents affecting both the central nervous system (CNS) and systemic mechanisms merge towards AD genesis; (iv). AD likely represents a complex condition displaying both central and peripheral peculiarities.

Specifically, we try to connect the loss of spiral ganglion neurons (SGNs) in the auditory system during physiological aging, causing sensorineural hearing loss to AD development. We give proof that ARHL can be considered a risk factor for cognitive decline and AD development. As an outlook, we discuss the innovative lines of research to inhibit the aggregation and deposition of Aβ more profitably in brain parenchyma through targeted neurotrophic factor supplementation to the cochlea; this is to prevent AD development.

## 2. The Blood–Brain Barrier and Its Dysfunction in the Genesis of AD

The blood–brain barrier (BBB) is formed by the cerebral microvascular endothelium, which, through its tight junctions, works to control the paracellular transport of hydrophilic and charged substances (Figure 1) [28,29]; this thereby guarantees the maintenance of homeostasis in the central nervous tissues [30,31,32]. The BBB acts as a strict control point for the regulation and differentiation of the CNS to control its energy supply and permeability.

Under inflammatory conditions and owing to the specialized structure of the BBB, immune cell entry into the CNS parenchyma involves two distinct, regulated steps: migration of immune cells across the BBB or blood–cerebrospinal fluid barrier (BCSFB) into the cerebrospinal fluid (CSF)-drained spaces of the CNS, followed by progression across the glia limitans into the CNS parenchyma. So far, research has focused mainly on the elucidation of the distinct molecular mechanisms required for immune cell migration across the different CNS barriers in the setting of multiple sclerosis [33], brain cancer [34] and ischemic brain injury [35]. Meanwhile, many other neuroinflammatory diseases, such as AD and its comorbidities, still await more profound mechanistic investigation.

## 3. Blood–Brain Barrier Dysfunction in the Pathogenesis of Alzheimer’s Disease—Available Knowledge

The impaired function of the BBB in AD leads to the impaired clearance of neurotoxic Aβ, which causes impaired cognitive function [36,37]. Moreover, cerebrovascular dysfunction is one of the key underlying factors of AD pathogenesis, resulting in cerebral hypoperfusion.

A dysfunctional BBB in AD patients has been shown by means of imaging studies and is detectable at multiple levels in AD [38]. In a subgroup of this patient cohort, even cerebral microbleeds were detected by MRI, thereby confirming a damaged cerebral vasculature [39,40,41]. Using PET technology, functional studies could assess BBB-related membrane transporter functions. The reduced activity of the global BBB efflux transporters P-gylcoprotein (P-gp) and ATP-binding cassette (ABC) was reported. P-gp has been characterized to be involved in Aß clearance, supporting the notion that impaired BBB function is a key contributing factor in the pathogenesis of AD [42,43]. Besides reduced membrane transporter activity, the altered expression of barrier-constituting proteins, such as tight junction proteins, leading to reduced BBB integrity, was reported as well [44]. The altered BBB function was, moreover, speculated to be involved in other manifestations of cerebrovascular dysfunction, i.e., vasospasm, perturbations in cerebral blood flow and cerebral blood coagulation [45]. The correlation between BBB dysfunction and these latter cerebrovascular disturbances remains to be clarified.

Although the cellular and molecular steps in the frame of the disease process still await elucidation, the effects of brain microvascular dysfunction seem to promote neuronal loss and/or degeneration. This is in addition to senile plaque (Aβ) and NFT (tau) pathology [10] in AD. The vascular hypothesis claims that events such as hypoxia or blood–brain barrier (BBB) dysfunction are associated with the enrichment of vasotoxic and neurotoxic compounds in the brain. This is supposed to be sufficient to induce neurodegeneration independently or prior to Aß deposition [11,12,13,46]. On the other hand, cerebrovascular/ BBB dysfunction might be chiefly responsible for insufficient Aß clearance from the brain [13,47]. A dysfunctional leaky BBB might promote an increased influx of peripheral Aβ from systemic amyloidosis, the contribution of which to AD development is so far still under debate [48]. Moreover, an elevated expression of β-APP [49,50] could result in further Aβ accumulation in the brain, chiefly reported to be situated in close proximity to cerebral blood vessels [51]. Reports have stated that Aß was detectable in the vessels outside the cranial for several hours, associated with albumin or red blood cells. Several hours later, Aβ was detected to be attached to the vessel walls [51]. Elevated levels of Aβ in brain parenchyma are, in turn, hypothesized to induce or promote cerebrovascular [48,52] and neuronal [53] dysfunction, leading to a cycle of self-propagation [54,55,56]. Similar to prion diseases [57], it ultimately leads to cerebral β-amyloidosis [58].

Several vascular risk factors, specifically their effects on BBB integrity and function, have been discussed in the recent literature to increase the predisposition to AD [29]. Multiple epidemiological studies have demonstrated a remarkable overlap among risk factors for cerebrovascular disorder and sporadic, late-onset AD [59]. For example, mid-life diabetes [60], hypertension [16,61], and obesity [62] have been shown to increase the risk for both AD and vascular dementia. It is now generally acknowledged that most AD cases have mixed vascular pathology and small-vessel disease [12,63]. Moreover, reduced brain blood perfusion [64], silent infarcts [65], and the presence of one or more infarctions [66] all increase the risk of AD.

Atherosclerosis in the carotid arteries supplying the brain increases the risk of dementias, most prominently AD, through plaques in the intimal medial thickness region, leading to accelerated and pronounced cognitive decline [42]. Besides leading to Aβ plaque formation, high plasma cholesterol levels have been discussed to promote Aβ formation in a feed-forward mechanism in the hippocampal neurons; this augments the induced neurodegeneration even more [67,68].

However, vice versa, it could be shown that in addition to AD, Aß may be a relevant pathological feature in atherosclerosis, potentially a marker of plaque vulnerability. Based on these findings, the hypothesis of Aβ deposit involvement in the pathophysiology of atherosclerosis was formulated; while they first appear as distinct diseases, the pathologies of AD and atherosclerosis have similarities not limited to plaque deposition. Very recent studies were able to show that elevated concentrations of soluble APP and Aβ in the bloodstream are associated with higher cardiovascular risk. It has been hypothesized that APP from platelets is further processed to Aβ aortic sections from atherosclerotic mouse models; this equally showed cell-rich plaques with lipids and fibrotic tissue.

Besides this, ischemic damage, produced through vascular occlusion, enhances BBB permeability, inflammation and oxidative stress; consequently, it promotes the development of dementia [69]:

Stroke: Altogether, a significant association between stroke and the development of AD has been acknowledged. As underlying mechanisms, the depletion of oxygen and nutrients and the increased formation of inflammatory mediators and ROS have been detected. Consequently, as potential predictive cues for ischemic stroke events, changes in cerebral blood flow and increased BBB permeability might be predictive for the progression of AD [45,67]. Ischemic conditions, however, do not only trigger the classical effects of stroke but have been associated with Aβ senile plaque and NFT formation. In summary, affected patients thus experience not only cerebrovascular functional impairment but also vascular dementia. Besides clinically manifest stroke events, many subclinical forms of cerebrovascular insults and insufficiency (micro-embolisms, microcirculatory changes, silent strokes, micro-embolism) might add up to a continuum of pathological events, ultimately cumulating in AD pathology [68,70].

Moreover, to be specific, conditions related to metabolic syndrome, such as mid-life diabetes [60,71,72,73] and obesity [62,74,75], and the resulting atherosclerosis [42,51,67,68,70] and hypertension [76,77,78] have been shown to increase the risk for both AD and vascular dementia:

Diabetes: The diagnosis of type 2 diabetes mellitus has been positively correlated with an increased risk of developing AD [71,72,73]. Animal experiments could reveal an increased and exacerbated cognitive impairment with increased insulin resistance; besides hyperinsulinemia, it is a hallmark of type 2 diabetes mellitus. As to potential pathophysiological reasons, increased mitochondrial damage in the course of the disease is hypothesized to amplify the production of cell-toxic ROS; this is under investigation [73,79].

Obesity: Obesity and metabolic syndrome lead to a significantly elevated risk of developing AD. The elevated plasma levels of cholesterol and free fatty acids are suspected to not only promote a surge in the production and deposition of tau and Aβ proteins but presumably pro-inflammatory cytokines as well. In contrast, they might lower the production or expression of the matrix metalloproteinases involved in clearance. A dysfunctional BBB, however, facilitates the accumulation of harmful neurotoxic endo- and xenobiotics, resulting in neuronal damage or death and contributing to cognitive decline [74,75].

Hypertension: Hypertension is a key risk factor for developing AD by inducing vascular changes that, in turn, promote atrophy; this increases the deposition of NFT and Aβ plaques [76,77]. Amongst these vascular changes, neuroimaging, post-mortem analyses and ultrastructural observations could delineate phenomena such as basement membrane thickening, the reduction or degeneration of pericytes stabilizing the capillaries of the BBB and even erythrocyte accumulation. In summary, this leads to increased BBB permeability and cognitive decline (rev. in: Chakraborty, A. [78]). In the mouse model, it could likewise be confirmed that hypertension increases Aβ accumulation in the brain [80]. As molecular triggers, apolipoprotein E4 has been determined to increase Aβ deposition. On the other hand, angiotensin and angiotensin convert an enzyme, ACE, that appears to cause vasoconstriction. This drives an increase in blood flow, which is thought to underlie vascular damage, e.g., by increased ROS production [81]. Other lines of research hypothesize that hypertension promotes cognitive decline through the development of white matter based on impaired vascular function.

## 4. Age-Related Hearing Loss in the Genesis of AD

ARHL or presbycusis is the third most prevalent chronic health affliction in the elderly; it has been shown to correlate positively with the risk of cognitive impairment and AD development at an advanced age [5]. Steadily increasing with age, ARHL affects more than 30% of individuals older than 65 years, about 65% of people over 70 years, and 80% of people over 85 years [5]. The major cause of ARHL is lifetime exposure to insults to the auditory system. Clinically, ARHL is a progressive, bilateral, and multifactorial disorder that affects hearing sensitivity. Progressive hearing impairment leads to comorbidities such as frailty, falls, social isolation, late-onset depression, cognitive decline and an increased risk of dementia. This eventually contributes to AD development. Based on this, the US National Institute on Aging rated ARHL as a possible risk factor for AD.

Recent findings from research into ARHL from both human and animal studies may help to make progress in the ongoing discussion about future prospects for advances in our understanding of genetic susceptibility and pathology. This could eventually facilitate potential therapeutic approaches in ARHL as a modifiable risk factor for AD development.

The National Institute on Aging and the Alzheimer’s Association have developed operational criteria for diagnosing the preclinical stages of AD. High-capacity neuroimaging tools can be employed for better clarification of the confounding variables of AD risk factors, such as age, gender, apolipoprotein E4 status and cardiovascular risk factors [82]. NFT is present in the primary auditory and auditory association cortices of AD patients [82,83]. If the novel neuroimaging tools elucidate the development of NFT in the peripheral auditory system of ARHL patients in the early, progressive phase of AD, it will provide evidence for ARHL being a modifiable risk factor. In conjunction with suitable treatment, this could improve the quality of life and slow down cognitive decline in the elderly [84].

Focusing on individuals showing cognitive impairment, recent studies have investigated the link between ARHL and mild cognitive impairment (MCI) or AD. The Adult Changes in Thought (ACT) trial demonstrated that ARHL is related to executive dysfunction in older people with and without memory impairment and dementia [85]. Similarly, diverse cross-sectional case–control studies and two population-based studies suggest a clear involvement of age-related central auditory processing disorder (CAPD) in MCI, dementia, and AD [86]. It was shown that potential physiological alterations in both the central and peripheral auditory systems contribute to ARHL [87,88]; affected individuals showed high-frequency hearing impairment (6000 and 8000 Hz), compromised localization of sound sources, and slowed central processing of acoustic information. Specifically, CAPD was defined as a specific deficit in the processing of auditory information along the central auditory nervous system, including bottom-up and top-down neural connectivity in one or more areas of auditory discrimination, temporal processing and binaural processing. Clinically, this phenomenon is characterized by difficulty in understanding words in background noise in aged individuals (rev. in Loughrey et al., 2018) [86]. To complicate its diagnosis, CAPD appears to have a highly heterogeneous clinical presentation, rendering it difficult to distinguish from a peripheral auditory deficit. Therefore, current recommendations underline the benefit of an assessment of both peripheral and central auditory dysfunctions to better understand the origin of ARHL and its diverse impact on cognitive disorders of aging in the future.

Recent evidence on both familial and sporadic AD could show that the disease has a long preclinical phase in both cases that is presumably triggered more than 20 years before the onset of clinical AD symptoms [89,90,91]. On the other hand, ARHL is already manifested. Neuropathological studies underline a potential key role of degeneration in the auditory pathway in the frame of AD pathogenesis from NFT formation; this is primarily as a hyperphosphorylated tau protein, which is reported in the central auditory pathway long before brain Aβ deposition is detected [87]. In contrast, Aβ deposition is uncommon in the central auditory pathway. The data suggest that neurodegeneration in the auditory system could be an ongoing process in the frame of the AD course [83]. We conclude from this that identifying the common mechanisms that define the underlying epidemiological link between ARHL and AD has significant implications. It is speculated that this association might be related to an age-dependent decline in the structure and function of the supplying cerebrovascular sections of the BBB and blood–labyrinth barrier (BLB), respectively. It places cerebrovascular decline at the root of both conditions, ARHL and AD.

On the other hand, ample evidence has been given in animal studies, demonstrating that the induction of hearing loss in animals produces cognitive decline, particularly hippocampal dysfunction, mainly due to elevated ROS and excitotoxicity. This, in turn, leads to the cessation of neurogenesis, synaptic loss and tau hyperphosphorylation. The experimental data suggest that hearing loss can lead to pathological hallmarks similar to those seen in AD and other dementias.

At a molecular level, tests in mice at 12 months of age showed that the noise-induced hearing loss mice exhibited higher levels of phosphorylated tau and larger lipofuscin granule areas (a marker of brain aging) [92,93,94] in the hippocampus. The finding that the working memory impairment was reversible but that the recognition memory dysfunction was permanent suggests that the effects of noise on different aspects of cognition are separable. However, the rodent data do not establish that hearing loss on its own can induce a progressive degenerative dementing illness. Therefore, we conclude that an additional “hit”, such as aging, APOE genotype, microvascular disease or others, may be necessary to trigger an ongoing degenerative process such as AD.

### 4.1. Functional Contribution of Age-Related Decline in BLB Integrity on ARHL

In search of a common determinant, it should be emphasized that the two aging-associated disorders, AD and ARHL, seem to share a common underlying pathogenesis. Based on the presented evidence, we argue that the identification of a common pathological target linking ARHL and AD could represent a new thread for the development of therapies for both disorders. This common underlying pathogenesis appears to be linked to age-related cerebrovascular dysfunction and the dysfunction of the BBB [78] and BLB [95]. They supply the brain and the cochlea of the inner ear, respectively. The BLB, located in the cochlear lateral wall, is the barrier between the vasculature and the inner ear fluids. It is critical for auditory and vestibular function by maintaining ion homeostasis and transporting nutrients and systemic hormones to the inner ear [95]. The cerebrovascular qualities and similarities of the BLB to the BBB have been described recently [96] (Figure 2).

While the blood vessels of the inner ear are critical for normal function, they are, however, at the same time, highly vulnerable to pathologic events. They result in hearing and vestibular dysfunction; a decrease in the area or volume of the stria vascularis [88], specifically, the integrity of the BLB in the cochlear lateral wall of rodents, for example, is the most commonly described finding with the increasing age of the animals. The underlying molecular pathological processes are, however, poorly understood. To enable preventive and curative care, aging-induced BLB dysfunction awaits comprehensive characterization [95,97].

In general, a common trigger for both AD and ARHL might lie in the aging-induced decline in BBB and BLB, respectively, with alterations at the BBB promoting the development of AD and the aging-related dysfunction of the BLB fostering ARHL. The preservation of the integrity of both cerebrovascular systems might help to prevent both AD and ARHL by the maintenance of the respective tissue homeostasis function. Comparably, common microvascular pathology has been proposed to be at the root of diabetes-associated complications involving the microvasculature of eyes, heart and kidneys, respectively [98,99,100]. Our hypothesis, linking AD genesis to ARHL, is further strengthened by the fact that many cerebrovascular diseases are associated with frailty; for example, stroke, congestive heart failure, and diabetes-associated complications seem to include cognitive impairment and disturbances in hearing as comorbidities [101].

### 4.2. The Age-Related Decline of Spiral Ganglion Neurons—Could the Treatment of ARHL Improve Cognitive Outcomes or Prevent AD Development?

Given the large social and welfare burden that results from both AD and ARHL and because ARHL is potentially a modifiable risk factor for dementia, there is an urgent need for therapeutic interventions to ameliorate age-related auditory decline caused by SGN degeneration [82,87]. However, a prerequisite for the design of drug-, gene- or cell-based therapies would be an understanding of the underlying molecular mechanisms [95,97,102]. Currently, our understanding of ARHL is very limited.

Sensorineural hearing loss, in general, can be caused by a loss of either hair cells, the sound-transducing sensory cells of the cochlea or the primary auditory neurons, the SGNs, which connect the hair cells to the cochlear nucleus in the brainstem. The age-related loss of SGNs is consistently observed in humans and animals. The degeneration of SGNs can occur secondarily to the loss of hair cells that maintain them through the supply of trophic support [103] or primarily before hair cell loss occurs.The loss of hair cells is hypothesized to be chiefly responsible for age-related SGN loss. In humans, SGN loss and the relative preservation of the organ of Corti is called auditory neuropathy [104]; if it is age-related, it is classified as neural presbycusis [105]. Not all SGN loss is debilitating or necessarily merits a label of neural presbycusis. Humans are born with a substantial surfeit of neurons that exceed the number necessary for sound detection about ten times but only twice the number necessary for fine frequency discrimination [106]. A meaningful label of neural presbycusis requires proof of an accelerated loss of SGNs that progressively impairs sound perception at a rate that would exceed the overall healthy ‘biological age’ of the individual. Early manifestations of this may include decreased speech intelligibility (specifically in noisy environments), poor signal-to-noise ratios, and impaired frequency resolution [105]. The latter problem might add to an impaired ability to identify or spatially localize natural sounds and likely alters the perceptual ability to appreciate music. The fact that not everyone exhibits neural presbycusis implies that both genetic and environmental factors contribute to this condition. These complicated interactions can be manifested at the systemic, organ, and molecular levels. Even though it is, of course, important to distinguish primary SGN loss from secondary degeneration, it is well understood that SGNs begin to die the moment that trophic support is not available any more [107,108,109,110].

To conclude, age-related loss of SGNs is a key component of ARHL. Evidence from recent investigations shows that the survival of SGNs during physiological aging depends on both genetic and epigenetic influences, which have been demonstrated at the systemic, tissue, cellular, and molecular levels [111]: Studies in mice have highlighted that the survival of SGNs is dependent on a number of genes encoding for neurotrophins [112]. At the cellular and molecular level, several pathways, particularly free radical and calcium signaling pathways, can promote age-related SGN loss. Future studies might determine how these pathways, contributing individually to SGN loss, act on SGNs directly or indirectly. with the advancement of recent genetic and pharmacologic tools, we should not only try to understand how SGNs die during aging but also find ways to delay or even prevent this loss.

The central axons of SGNs bundle together to form the modiolar segment of the auditory nerve. SGN activation is processed via the auditory nerve, which transmits auditory information up a series of nuclei to the auditory cortex, where perception occurs. It is well known that hearing loss affects various regions of the central auditory pathway [113,114], leading to cytoskeletal pathology and functional deficits [115,116]. However, only limited studies have tried to elucidate how those hearing-loss-related central changes correlate to AD development. Central auditory processing tests are behavioral tests that are difficult to perform in people with AD, which could explain the limited number of studies with AD participants that look at central auditory processing using behavioral tests [117]. Having reviewed 18 studies on the central auditory processing function in AD and its preclinical stages, Tarawneh and colleagues concluded that a specific test battery can be used as a hearing biomarker/clinical tool to early-identify older adults at risk of cognitive impairment in clinical settings [117].

Whether the treatment of ARHL can preserve cognitive function in the elderly has been addressed directly and indirectly in several studies (rev., e.g., in Amieva and Ourvard, 2020 [118]). The epidemiological data presented underline that hearing aids positively impact long-term cognition even though interventional studies are lacking so far. However, more profound research will be necessary to refine the statement with respect to the length or frequency of hearing aid use by elderly individuals enrolled in the study. Information on the benefits of cochlear implants (CI), on the other hand, is more scarce [119]. Although CIs can effectively replace the mechanosensory transduction function of lost hair cells by providing direct electrical stimulation of SGNs, this technique can only be successful as long as sufficient residual SGNs remain [120]. Gathering such information will be important as CI users experience even more severe hearing impairments that have yet to be correlated with AD development or prevention.

## 5. Neurotrophic Factors at the Interface of AD Pathophysiology and Neural ARHL

Neurotrophic factors are a group of secreted proteins that have multiple functions in neural and non-neural tissues, e.g., mediating the development, homeostasis and survival of the central and peripheral nervous systems [121].

An imbalance of neurotrophic factors and their receptors has been reported for both AD and ARHL. In this context, nerve growth factor (NGF), GDNF and BDNF stand out [122,123,124,125,126,127,128]. The identified neurotrophic factors affected have been shown to have meaningful implications in the frame of neuronal degeneration processes. In both disorders, these neurotrophins are suitable for preventing cell death and enhancing growth and function. They thus seem to play an essential role in the survival of neurons affected by the degenerative processes in ARHL and AD, respectively. The neurotrophins NGF, BDNF and GDNF and their receptors are expressed in the inner ear and the CNS and are affected by age, as detailed in these articles [129,130,131].

Since both diseases are affected by neurotrophic factor imbalance and therapy as well, the already well-known hypothesis that ARHL might be at the root of AD genesis may be true. Another hypothesis speculates that the age-related imbalance of these neurotrophins may be a link between AD and ARHL. In other words, hearing loss does not promote AD, but rather, a fundamental common pathology—such as cerebrovascular dysfunction—causes both disorders.

The hen-and-egg causality question, i.e., which pathology subsequently affects the other, may, however, be irrelevant in view of the therapeutic effect of the neurotrophic factor therapy. The supply of the aforementioned neurotrophins reduces the anatomical correlates of neuronal ARHL by preserving SGNs from degeneration. as a consequence, a potent therapeutic option in the preclinical models of AD and neural ARHL could emerge to treat both diseases at the same time. These therapies appear to be promising to protect or promote the regeneration of both hair cells and SGNs, thus regenerating or even preventing deafness of the mammalian inner ear [130]; meanwhile, they positively affect AD. However, their safety and efficacy in clinical studies are not yet proven [125,131,132]. Further future research and randomized clinical trials are warranted to examine and foster these implications of treatment for cognition and ARHL on a more profound level. This would facilitate the exploration of possible causal mechanisms underlying their kinship.

## 6. Current and Novel Treatment Modalities

Current pharmaceutical approaches targeting neuropathologic processes such as AD offer limited, predominantly symptom-modifying effects [132,133] but have failed so far to prevent further neurodegeneration. The present available therapeutic targets are limited to two candidates that were shown to improve cognitive function, i.e., acetylcholinesterase (AChE) inhibitors and N-methyl-D-aspartate (NMDA) receptor antagonists.

## 7. Novel Inhibitors of Amyloid Fibril Formation in AD

The pathological process in AD is based on the formation of amyloid fibrils (Aβ) and amyloid oligomers, which have to be explored more efficiently; they encourage the inhibition of the Aβ aggregation process [134,135]. Therefore, by developing and applying the appropriate techniques for measuring the Aβ affinity to drug-like and supramolecular structures in the solution, it is possible to discover novel therapeutics against AD. Previously, the amyloid β-sheet breaker was determined as N,N’-bis(3-hydroxyphenyl)pyridazine-3,6-diamine (RS-0406), capable of significantly inhibiting 25 mM Aβ fibrillogenesis (Figure 3A,B). This would suggest that RS-0406 or one of the derivatives could become a therapeutic agent for AD patients [136].

Moreover, atomic force microscopy, together with molecular dynamics simulations, has demonstrated that some pseudo-peptides might bind to amyloid fragments with different affinity; this would prevent the Aβ-Aβ aggregation process [137]. Similarly, other β-sheet breaker peptides have been designed to complement the enthalpic interactions with the aggregating protein and inhibit the pathological process in a concentration-dependent manner [138]. Additionally, the neuroprotective effect of single-wall carbon nanotubes with built-in peroxidase-like activity against β-amyloid-induced neurotoxicity was established to diminish the formation of Aβ fibrils [139].

On the other hand, cyclodextrin-based formulations of different drug-like molecules have also been developed in many previous studies [140,141]. Some modified cyclodextrins, such as hydroxypropyl-beta-CD (HPβCD), have already been implemented as inhibitors in cell and mouse models of AD [142,143]. In particular, it has been shown that the subcutaneous administration of HP-β-CD to Tg19959 mice significantly improved memory deficits and reduced amyloid deposition, microgliosis, and immunoreactive dystrophic neurites. These effects occurred, at least in part, by reducing the amyloidogenic processing of APP and enhancing ABCA1-mediated Aβ clearance. The present data suggest that HP-β-CD may have therapeutic potential for the treatment of AD, but more research is needed to develop highly efficient and cytotoxicity-free excipients with improved pharmacokinetics and pharmacodynamics (Figure 4).

In a transgenic mouse model of AD, a monoclonal antibody, aducanumab, was shown to selectively target Aβ plaques [144]. This compound could enter the brain, binding parenchymal Aβ, followed by the reduction of its soluble and insoluble variants in a dose-dependent manner [144]. In summary, compounds such as peptide inhibitors, carbon allotropes, biologicals (antibodies), small molecules and supramolecular complexes that target specific Aβ subregions represent the first generation of amyloid-based therapeutics with the potential to demonstrate disease-modifying activity. Gaining additional insights into amyloid biology and AD will likely guide the development of the next generation of inhibitors.

**Figure 3 biomolecules-12-01717-f003:**
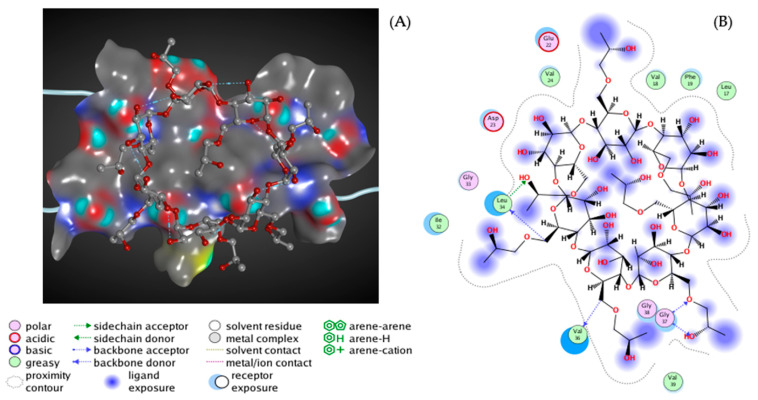
The predicted interactions of the RS-0406 molecule with Aβ as a trimer in 3D (**A**) and 2D (**B**). The trimer is depicted as a ribbon diagram, and RS-0406 is shown as a ball-and-stick model. The molecular surface represents the interaction between protein and ligand structures [145].

**Figure 4 biomolecules-12-01717-f004:**
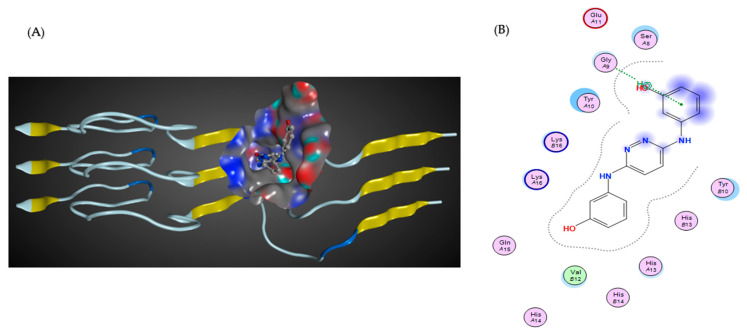
The predicted interactions of the HPβCD molecule with Aβ as a monomer in 3D (**A**) and 2D (**B**). The monomer is depicted in white color, and the cyclodextrin is shown as a ball-and-stick model. The molecular surface represents the interaction between protein and ligand structures. The predicted model was generated using the standard methodology (Lamarckian genetic algorithm), published elsewhere [146].

## 8. Summary and Outlook

In our review, we highlight the central role of cerebrovascular dysfunction in the genesis of AD and ARHL, respectively, and present potentially interrelated mechanisms that may contribute to their pathogenesis; this is likely to occur in a cascade of events at the brain–ear axis (Figure 5).

As the literature gives evidence that cerebrovascular dysfunction in the brain and the cochlea might accelerate both ARHL and AD pathologies, future investigations should address the respective preventive and therapeutic approaches. Due to the interconnectivity between ARHL and AD, the optimal therapeutic approach should not focus only on a single pathway. We propose that multimodal therapies are needed, aiming at a variety of specific pathogenic pathways at the same time, leading to synergistic results. We propose that such a multimodal therapeutical approach needs to include the targeting of the observed microvascular integrity and functional disturbances. Additionally, recent evidence indicates that ARHL might be a modifiable risk factor for AD development, with widespread epidemiological impact, and that the underlying neural presbycusis might be preventable by targeted neurotrophic administration to the cochlea. Suggestions to preclude ARHL and AD through the timely treatment of SGN loss by neurotrophic supplementation can be regarded as a key topic for the development of treatments for AD.

## Figures and Tables

**Figure 1 biomolecules-12-01717-f001:**
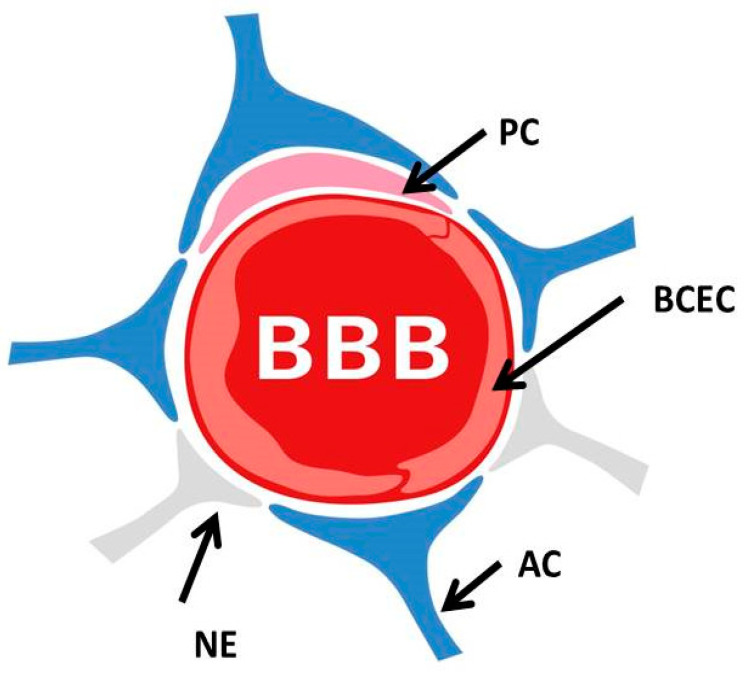
The blood–brain barrier. The blood–brain barrier is formed by endothelial cells of the capillary wall (BCEC: brain capillary endothelial cells), astrocyte (AC) end-feet surrounding the capillary and pericytes (PC) embedded in the capillary basement membrane. This system allows the selective passage of nutrients while restricting the entry of potentially neurotoxic compounds. NE: neuronal end-feet.

**Figure 2 biomolecules-12-01717-f002:**
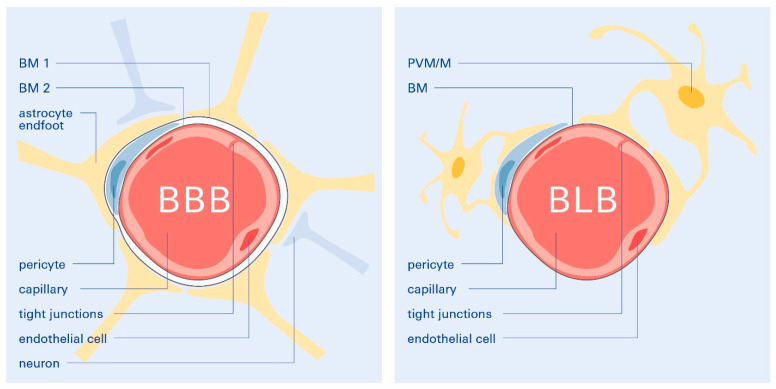
Illustration of the capillaries and cell types forming the BBB and BLB (cross-sections). The BBB endothelium is embedded in a double basement membrane while, at the same time, accommodating the supporting pericytes. Within the CNS, BBB capillaries are covered by astrocytic end-feet. The BLB endothelium is comparably surrounded by a basement membrane with pericytes ensheathing a simple basement membrane enclosing the capillaries. Perivascular resident macrophage-like monocytes (PVM/Ms) coat the basement membrane surrounding the BLB endothelial cells. In both the BLB and BBB, tight junctions are present between the endothelial cells occulting the paravascular space. In BLB and BBB, the tight junctions are composed of similar proteins.

**Figure 5 biomolecules-12-01717-f005:**
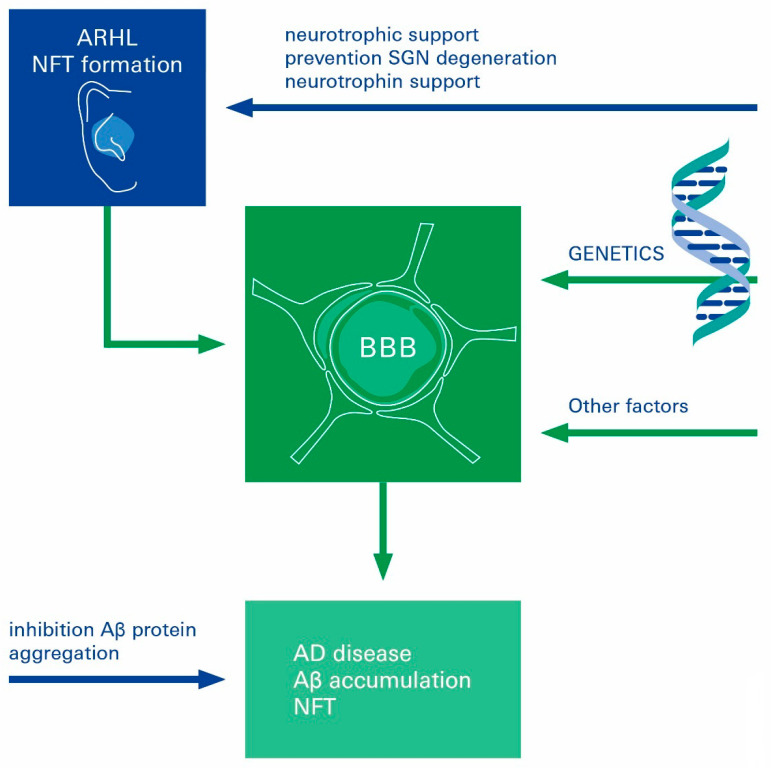
The brain–ear axis linking ARHL to the development of AD. Long before the diagnosis of AD symptoms, AD patients present with clinical signs of a functional defect in the auditory system, and tauopathy is present in the auditory tract. In the era of the epidemiology of increasing longevity, these findings suggest an important new possibility for the early detection and prevention of AD, as the maladies AD and ARHL may coexist and even share a common etiology, rendering AD a systemic disease. Potential intervention points for prevention or therapy by the inhibition of Aβ protein aggregation or neurotrophic supplementation to nurture and preserve spiral ganglion neurons are indicated by arrows. Aβ: amyloid-beta; AD: Alzheimer’s disease; ARHL: age-related hearing loss; BBB: blood–brain barrier, SGN: spiral ganglion neurons.

## Data Availability

Not applicable.

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
