# Peer review of "Linking Cerebrovascular Dysfunction to Age-Related Hearing Loss and Alzheimer’s Disease—Are Systemic Approaches for Diagnosis and Therapy Required?"

_biomolecules, 2022, doi:10.3390/biom12111717_

Round 1

Reviewer 1 Report

The review article by Forster et al. is focused on the role of the neurovascular dysfunction in ARHL and Alzheimer’s disease. Neurovascular system is essential for both auditory perception and central nervous system, therefore the topic of this review is important. The manuscript references most of the relevant recent literature, which is a strength of this work. However, in this manuscript, the authors also included several unpublished data, Fig. 3 and Fig. 4. The reviewer strongly recommends that these data be omitted unless they can provide essential details about how these predictions were performed, such as the type of software used etc.  

Author Response

The review article by Forster et al. is focused on the role of the neurovascular dysfunction in ARHL and Alzheimer’s disease. Neurovascular system is essential for both auditory perception and central nervous system, therefore the topic of this review is important. The manuscript references most of the relevant recent literature, which is a strength of this work. However, in this manuscript, the authors also included several unpublished data, Fig. 3 and Fig. 4. The reviewer strongly recommends that these data be omitted unless they can provide essential details about how these predictions were performed, such as the type of software used etc.  

A. Many thanks for this important request which will help to improve over the quality of the manuscript.

We added the requested information:

Figure 3. The predicted interactions of the RS-0406 molecule with Aβ as a trimer in 3D (A) and 2D (B). The trimer is depicted as a ribbon diagram and RS-0406 is shown as a ball-and-stick model. The molecular surface represents the interaction between protein and ligand structures (Shityakov et al., 2021).

Shityakov, S., et al. (2021). "The Conspicuous Link between Ear, Brain and Heart-Could Neurotrophin-Treatment of Age-Related Hearing Loss Help Prevent Alzheimer's Disease and Associated Amyloid Cardiomyopathy?" Biomolecules 11(6).

Figure 4. The predicted interactions of the HPβCD molecule with Aβ as a monomer in 3D (a) and 2D (b). The monomer is depicted in white color and the cyclodextrin is shown as a ball-and-stick model. The molecular surface represents the interaction between protein and ligand structures. The predicted model was generated by using the standard methodology (Lamarckian genetic algorithm) published elsewhere (Shityakov et al., 2016; Shityakov et al., 2021). 

Shityakov, S., et al. (2016). "Evaluation of the potential toxicity of unmodified and modified cyclodextrins on murine blood-brain barrier endothelial cells." J Toxicol Sci 41(2): 175-184.

Shityakov, S., et al. (2021). "Scaffold Searching of FDA and EMA-Approved Drugs Identifies Lead Candidates for Drug Repurposing in Alzheimer's Disease." Front Chem 9: 736509.

Reviewer 2 Report

This is a very good review paper; I have minor revision suggestions as follow:

Page 2, line 76-84. It would be adequate to cite the papers by Shen et al " Sensorineural hearing loss may lead to dementia-related pathological changes in hippocampal neurons" (Neurobiol Dis. 2021 Aug; 156:105408) and Tai et al " Association of sudden sensorineurial hearing loss with dementia: a nationwide cohort study" (BMC Neurol. 2021 Feb 25;21(1):88).    Citing their papers would make your opinion about the links among ARHL, cognitive decline and dementia strong.                                                                                  

Page3, line 99: For sentence " Our focus is to establish a link between the loss...", it would be better as " Our focus is to summarize key research of establishing a link between the loss..."

Page 6, line 268-274, it would be better to add reference 74 (Nadhimi et al. 2021) for this section.

Page 8, 9, Section 3.2, I think that this section provided with SGN as a highlight represents an excellent framework for this article, but I do believe that there is some merit to benefit from some extensive description from SGN to how Alzheimer's disease affects the brain (focus on the central auditory pathways and damages that many neurons stop functioning). Also, it would better to have an echo (central nervous system in addition to SGN) in the summary and outlook logically.

Author Response

This is a very good review paper; I have minor revision suggestions as follow:

Page 2, line 76-84. It would be adequate to cite the papers by Shen et al " Sensorineural hearing loss may lead to dementia-related pathological changes in hippocampal neurons" (Neurobiol Dis. 2021 Aug; 156:105408) and Tai et al " Association of sudden sensorineurial hearing loss with dementia: a nationwide cohort study" (BMC Neurol. 2021 Feb 25;21(1):88).    Citing their papers would make your opinion about the links among ARHL, cognitive decline and dementia strong.                                               

A. Thanks, we added the named citations.                                   

Page3, line 99: For sentence " Our focus is to establish a link between the loss...", it would be better as " Our focus is to summarize key research of establishing a link between the loss..."

A. thanks, the sentence was rephrased „Our focus is to summarize key research of establishing a link between the loss and AD development by highlighting..”

Page 6, line 268-274, it would be better to add reference 74 (Nadhimi et al. 2021) for this section.

A. thanks, Reference was added.

Page 8, 9, Section 3.2, I think that this section provided with SGN as a highlight represents an excellent framework for this article, but I do believe that there is some merit to benefit from some extensive description from SGN to how Alzheimer's disease affects the brain (focus on the central auditory pathways and damages that many neurons stop functioning). Also, it would better to have an echo (central nervous system in addition to SGN) in the summary and outlook logically.

A. Thank you for proposing this. We agree that the central auditory pathway plays a significant role in the SGN-ARHL-AD axis and  added details on this in lines 395 to 408. Additionally, we included the central pathway aspect in the outlook section, lines 513 to 515.

Reviewer 3 Report

This paper reviews aspects of the literature underpinning the mechanisms underlying Alzheimer's disease (AD) and age related hearing loss (ARHL) to identify neurovascular dysfunction as a common mechanistic entity that links the two conditions.

The work is in an area of need, as dementia and hearing loss have a significant burden of disease across much of the globe. The authors propose that the systemic nature of the conditions may make them amenable to disease modifying treatment through interventions that protect the blood brain barrier and blood labyrinth barrier. They suggest that neurotrophins may be a viable treatment, with the protection against the progression of hearing loss being their key target.

The paper is very poorly written and in need of significant revision and re-writing. The evidence that is presented is disjointed, and the thread of evidence that the authors are trying to present is fragmented and lacks a clear rationale.

Rather than trying to capture multiple aspects of the mechanistic underpinnings of Alzheimer's disease which span the roles of amyloid beta, tau and lifestyle factors a summary schematic showing the points of intersection and overlap between the different factors and mechanisms for both AD and ARHL and then drawing out the rationale for the hypothesis that the authors seem to be proposing would make the manuscript useful and help bolster the points the authors are raising. Consider a sharper focus on the vascular link, which appears to be central to the argument. There is a vast literature on AD, including some key recent review articles, consider reference to these rather than trying to generate a synopsis of AD mechanisms and treatment rationales then draw in the key new thinking which is how this links to ARHL.

Aspects of the terminology are not clear - what do the authors mean by neurovascular? 

lines 23-130 - it is not clear what point this is trying to convey

there are numerous syntax errors, including missing words/letters at points.

the references are not correctly numbered

Author Response

Consider a sharper focus on the vascular link, which appears to be central to the argument. There is a vast literature on AD, including some key recent review articles, consider reference to these rather than trying to generate a synopsis of AD mechanisms and treatment rationales then draw in the key new thinking which is how this links to ARHL.

  • thanks, in line 111 ff we wrote in a more distinct approach Vascular damage is believed by many authors to play a crucial role in the pathogenesis of AD in AD being primarily vascular rather than a neurodegenerative disorder (Attems and Jellinger, 2014). This hypothesis is mainly based on the ability of cerebral hypoperfusion caused by cortical microinfarcts to exacerbate a cognitive decline in AD (Miklossy, 2003). On the other hand, many epidemiological studies have shown the link between ARHL and AD associated with the loss of the brainstem and cerebellar volume (Llano et al., 2021). Moreover, this biomarker has been established for various cardiovascular and cerebrovascular pathologies, including congenital heart disease, stroke, and carotid stenosis (Marelli et al., 2016; Bu et al., 2021; Ammirati et al., 2020). In our review, we present AD as a systemic malady that may be caused by ARHL-related vascular decline in BLB integrity and function and following damage to spiral ganglion neurons. Our focus is to summarize key research on the establishment of a link between hearing loss and AD development..”

Attems, J. and K. A. Jellinger (2014). "The overlap between vascular disease and Alzheimer's disease--lessons from pathology." BMC Med 12: 206.

Llano, D. A., et al. (2021). "Reported Hearing Loss in Alzheimer's Disease Is Associated With Loss of Brainstem and Cerebellar Volume." Front Hum Neurosci 15: 739754.

Miklossy, J. (2003). "Cerebral hypoperfusion induces cortical watershed microinfarcts which may further aggravate cognitive decline in Alzheimer's disease." Neurol Res 25(6): 605-610.

Marelli, A., et al. (2016). "Brain in Congenital Heart Disease Across the Lifespan: The Cumulative Burden of Injury." Circulation 133(20): 1951-1962.

Bu, N., et al. (2021). "Early Brain Volume Changes After Stroke: Subgroup Analysis From the AXIS-2 Trial." Front Neurol 12: 747343.

Ammirati, E., et al. (2020). "Extent and characteristics of carotid plaques and brain parenchymal loss in asymptomatic patients with no indication for revascularization." Int J Cardiol Heart Vasc 30: 100619.

Aspects of the terminology are not clear - what do the authors mean by neurovascular? 

>A. thanks, we replaced the term “neurovascular” throughout by “cerebrovascular” which is pathomorphologically more correct.

lines 23-130 - it is not clear what point this is trying to convey

>A. 1. Thank you for this critical comment. In this section of the text, we allow ourselves to contrast the current theories on the ontogenesis of AD with the relatively recent notion that neurovascular dysfunction and in particular ARHL represent a key trigger for the hitherto incurable disease.

  1. Since our review submission “Linking cerebrovascular dysfunction to age-related hearing loss and Alzheimer’s disease– are systemic approaches for diagnosis and therapy required?“ is designed to be a contribution to the special issue "regulation of the endothelial barrier", it cannot be assumed that all vascular specialists who will frequent this editorial are also specialists in the field of neurodegenerative diseases. We therefore consider a generally understandable prelude to be justified and appropriate.

there are numerous syntax errors, including missing words/letters at points.

> Thanks for pointing this out. We apologize for the inadequate presentation and have screened the text throughout.

the references are not correctly numbered

> An update of citations and bibliography was performed using EndNote reference managing program.